# Ammonia and Nematode Ascaroside Are Synergistic in Trap Formation in *Arthrobotrys oligospora*

**DOI:** 10.3390/pathogens12091114

**Published:** 2023-08-31

**Authors:** Jinrong Huang, Xi Zheng, Mengqing Tian, Keqin Zhang

**Affiliations:** 1State Key Laboratory for Conservation and Utilization of Bio-Resources in Yunnan, Yunnan University, Kunming 650091, China; huangjinrong@mail.ynu.edu.cn (J.H.); zhengxi1@mail.ynu.edu.cn (X.Z.); 2Key Laboratory for Potato Biology of Yunnan Province, The CAAS-YNNU-YINMORE Joint Academy of Potato Science, Yunnan Normal University, Kunming 650091, China; 15198961953@163.com

**Keywords:** *Arthrobotrys oligospora*, ammonia, Acsr#18, Amt43, trap formation, synergy

## Abstract

Nematode-trapping (NT) fungi are natural predators of the soil living nematodes. Diverse external signals mediate the generation of predatory devices of NT fungi. Among these, broad ascarosides and nitrogenous ammonia are highly efficient inducers for trap structure initiation. However, the overlay effect of ammonia and ascaroside on the trap morphogenesis remains unclear. This study demonstrated that the combination of nitrogenous substances with nematode-derived ascarosides led to higher trap production compared to the single inducing cues; notably, ammonia and Ascr#18 had the most synergistic effect on the trap in *A. oligospora*. Further, the deletion of ammonia transceptor Amt43 blocked trap formation against ammonia addition in *A. oligospora* but not for the ascaroside Ascr#18 induction. Moreover, ammonia addition could promote plasma endocytosis in the process of trap formation. In contrast, ascaroside addition would facilitate the stability of intracellular organization away from endocytosis. Therefore, there is a synergistic effect on trap induction from different nitrogenous and ascaroside signals.

## 1. Introduction

Morphological transition is an elaborate adaptation strategy developed by many fungi to survive in different niches. These fungi have the ability to change their morphology when exposed to complex external stimuli. There are several invasion structures presented here as examples, such as “trap”, “appressorium”, and “yeast to hyphal”, all of which can contribute to infection or extension of organisms or parasites [1,2,3,4]. Membrane-bound transporters act as gatekeepers that control the import of external substances (e.g., sugars, amino acids, nucleobases, carboxylic acids, and metal ions) into metabolic/signaling pathways [5]. Specifically, the transporter GAP1 is responsible for general amino acids transport [6], the Amts for ammonia/H^+^ or ammonium [7], the SWEET16 for sugars [8], the Snf3 and Rgt2g for glucose [9], etc. Besides nutrition acquisition, plasma membrane (PM) transporters are still involved in the regulation of cellular morphogenesis [10]. For example, high-affinity glucose transceptor Hgt4 is required for the yeast-to-hyphal morphological switch in *C. albicans* or other yeast microorganisms, and ammonium transporters have an effect on appressoria formation in the filamentous fungi *Colletotrichum gloeosporioides* [11]. Hence, the PM transporter system is directly linked to diverse functions related to life history and pathogenicity in fungi.

As a preferred nitrogen source, ammonia or ammonium presents an important signal in fungal physiology or morphological transition [12]. Ammonium sequestration usually occurs at the periplasmic face, where NH_4_^+^ can be dissociated into neutral ammonia gas (NH_3_) and positively charged hydrogen (H^+^). Subsequently, the ammonium transporter/methyl ammonium permease (Amt/Mep) is responsible for the ammonia transport in and/or out of the cell [13]. Ammonium transporters are relevant to numerous forms of life; for example, ammonium utilization can improve crop yields [14], ammonium transporters in humans are essential for kidney function, and the related mutations have been implicated in various diseases [15]. Following sequestration of NH_4_^+^ at the periplasmic face, NH_4_^+^ is deprotonated and neutral NH_3_ is transported into the cytoplasm. The yeast *Saccharomyces cerevisiae* ammonium transport system comprises a network of Mep1, Mep2, and Mep3. Mep2 protein has the highest affinity, about twenty times higher than that of Mep1, for the uptake of both charged and uncharged NH_3_ and NH_4_^+^. Mep3 permease has the lowest affinity [16]. Under ammonium limitation, Mep2 acts as an ammonium sensor, generating a signal that leads to pseudohyphal differentiation that has been reported in a variety of fungi including *Aspergillus nidulans*, *Cryptococcus neoformans*, *Candida albicans*, and *Ustilago maydis* [17,18,19,20]. Other than that, ammonium uptake and release can induce appressorium formation during host penetration by triggering signaling pathways in the entomopathogenic fungus *Metarhizium robertsii* as well as in the fungal pathogens *Colletotrichum gloeosporioides* and *Alternaria alternate* [21].

The Amt/Mep/Rh family of integral membrane proteins comprises ammonium transporters among eukaryotes. Having more than one Mep paralogue with high conservation is typically common in a variety of species from bacteria, archaea, fungi, plants, to animals [22]. All known Mep/Amt permeases are 400–450 amino acids long and are predicted to have 10–12 transmembrane (TM) helices with Nout–Cin topology, as well as a central channel for the transport of ammonium [23,24]. About the flow of how ammonium is transported in and out of cells, the most recent research demonstrated that NH_4_^+^ dissociates into an ammonia gas (NH_3_) and a positively charged hydrogen (H^+^), which then traverse the cell membrane separately, and once inside the cell, NH_3_ and H^+^ recombine to make ammonium (NH_4_^+^) [25]. Even though there is a surplus of ammonia transporters in each organism, usually one of them serves as the transceptor that is responsible for transporting and sensing for ammonium/ammonia. Taking the yeast Mep system as an example, only Mep2 is defined as the transceptor, which functions not only as a transporter but also as a receptor during the nutrient-sensing process to activate the downstream signaling pathways [26]. Previously, it was demonstrated that urea and its downstream metabolite ammonia can trigger downstream signals via Amt transporter proteins and shift the lifestyle of nematode-trapping (NT) fungus *Arthrobotrys oligospora* from saprophytic mycelium to an adhesive predatory network [27].

Nematode-trapping fungi are a large family that hunt nematodes by producing traps, a property that has important potential for bio-controlling nematodes [28,29,30]. Amongst them, *A. oligospora* is commonly used as a model species for studying the interaction between NT fungi and nematodes; this fungus possesses a nutritive mycelium specialized in forming adhesive nets when nutrition is limited or in the presence of nematodes, and it then captures nematodes to obtain nutrition. Therefore, trap formation is an important survival strategy for this type of fungus to adapt to complex habitats and a key biological factor to determine the population size of nematodes in soil [31,32].

The nematode secretes a class of small molecule substances, ascarosides, that can trigger NT fungi to produce traps when interacting with NT fungi [33]. At present, around two-hundred types of ascarosides have been identified, and many of them overlap in their effects. For example, Ascr#2 and Ascr#5 are components of the pheromone that induces larval stagnation (dauer), and Ascr#2, Ascr#3, and Ascr#8 are used as effective male attractants [34,35]. In the present study, it was found that the addition of ammonia promotes higher trap formation during the action of ascarosides on *A. oligospora*. Through further analysis of the ammonia transporter Amt43 on trap formation and fungal pathogenicity to nematodes, the synergistic effect of ammonia in the reception of ascarosides signals by *A. oligospora* were explored.

## 2. Materials and Methods

### 2.1. Fungal Strains and Culture Conditions

The *Arthrobotrys oligospora* ATCC24927 strain was grown on potato dextrose agar (PDA) medium at 28 °C and stored in the State Key Laboratory for Conservation and Utilization of Bio-Resources in Yunnan. *Escherichia coli* strain DH5α (TaKaRa Bio, Dalian, China) was cultured on LB medium (0.5% yeast extract, 1% tryptone, 1% NaCl) as a host for plasmid *pCT74*. Additionally, CMY medium (3% maize boiled for 30 min, 0.5% yeast extract, and 1.5% agar) was applied to culture spores of *A. oligospora* and the mutants; WA medium (deionized water with 1.5% agar), CM medium (2% maize boiled for 30 min) for trap activity induction experiments, liquid TG medium (1% glucose,1% tryptone), TYGA medium (1% tryptone, 0.5% yeast extract, 1% glucose, and 1.5% agar), and PDA were used for the cultivation and observation of wild type (WT) and the mutants; LB medium was used to culture *E. coli*; and PDAS medium (PDA supplemented with 0.4 M sucrose and 10 g/L molasses) was used for protoplast regeneration of *A. oligospora*.

### 2.2. Deletion of the Gene Amt43

According to conservative informatics analysis, *Amt4*3 (AOL_s00043g163) may be the major ammonia transporter coding gene in *A. oligospora.* To analyze the gene function of *Amt43*, *Amt43* was deleted through homologous recombination (Appendix A) [36]. Briefly, the *Amt43-*5′ and *Amt43-*3′ flanking regions corresponding to each open reading frame were amplified using primers, and primers *Hph*-F and *Hph*-R were used to amplify the *HPH* cassette from the plasmid *pSCN44* (Appendix A). Then, the 5′ and 3′ flanking sequences of the target genes as well as the cloned *HPH* fragment were subcloned into the linearized *pRS426* vector (digested with *EcoRI* and *XhoI*) and then co-transfected into yeast strain FY834 by electroporation. The construct plasmid *pRS426*-*AoAmt43*-*HPH* was kept in *E. coli* DH5a. Finally, the constructed vectors were transformed into *A. oligospora* protoplasts, and Δ*AoAmt43* mutant strains were obtained by hygromycin B (200 μg/μL) resistance screening [37].

### 2.3. Southern Blot Analysis

Positive transformants were analyzed by Southern Blot. The hybridization probe was amplified using primers *Amt43*ko-F/*Amt43*ko-R, restriction enzyme *Mef*I was used to digest the genomic DNA of *A. oligospora*, and experiments were carried out according to the instructions provided in the North2South^®^ Chemiluminescent Hybridisation and Detection Kit (Meridian Rd., Rockford, IL, USA). Subsequently, the samples were washed with 0.25 M HCl for 13–15 min, rinsed with ddH_2_O for 15 min (repeated 3 times), treated with denaturing solution (NaOH 0.5 M, NaCl 1.5 M) for 15 min (repeated 2 times), rinsed in ddH_2_O for 15 min, treated with neutralizing solution (Tris-HCl 0.5 M, NaCl 1.5 M, PH = 7.5) for 15 min (repeated 2 times), rinsed in ddH_2_O for 15 min, and equilibrated in 20× SSC for 10 min, while the nylon membrane was soaked in ddH_2_O for 5 min and then transferred to equilibrate in 20× SSC for 30 min, and salt bridges were constructed for transferring membranes for 30 h. After the membrane transfer, the nylon membrane was cleaned with 6× SSC, blown dry with cold air, and then cross-linked with 1200 μJ/cm^2^ UV, and hybridization, membrane washing, and imaging finally confirmed that the positive transformants obtained were the correct mutants.

### 2.4. Amt43 GFP Labeling and Amt43 Overexpression

To determine the localization of Amt43 in *A. oligospora*, the gene was fused with GFP. Using *A. oligospora* genomic DNA as a template, three fragments of gene promoter, gene ORF, and terminator were amplified with the primer pairs *Amt43*P-F/*Amt43*P-R, *Amt43*-F/*Amt43*-R, and *Amt43*T-F/*Amt43*T-R. Subsequently, they were assembled into the *pCT74* plasmid including the GFP fragment (digested by *EcoR*I and *BsrG*I) to obtain a plasmid containing the promoter-Amt43-GFP-terminator. Then, the plasmid was transformed into *A. oligospora* protoplasts, and colonies were selected on PDA medium containing hygromycin B (200 μg/mL). After the colonies grew for 7 days, they were transferred to WA medium with cellophane on the surface and incubated at 28 °C for 3–4 days, and the cellophane with mycelium areas (0.5 × 0.5 cm) was taken and placed under the fluorescence microscope to observe whether the mycelium could emit fluorescence [36]. Similarly, the plasmid containing the Amt43::GFP was converted into the protoplasts of the Δ*AoAmt43* strain, and colonies were selected on PDA medium containing nourseothricin sulfate (100 mg/mL) to obtain the overexpressed strain Δ*AoAmt43::Amt43*.

### 2.5. Endocytosis Analysis

In order to investigate the difference in endocytosis between WT and mutant strains, spores from WT and mutant strains were inoculated on a cellophane-coated WA medium and incubated at 28 °C for 3 days. The cellophane layer of the medium was cut, stained with FM4-64 of 5 μg/mL (Biotium, Fremont, CA, USA) for 1 min or 5 min, rinsed three times with ddH_2_O, then fixed onto slides, and the stained samples were observed under a fluorescence microscope [38].

### 2.6. Comparison of Strain Growth

To assess the growth differences between WT and mutant strains, the two strains were first cultured for adaptation on PDA medium for 7 days at 28 °C. Then, we collected the same size (6 × 6 mm) blocks and inoculated them onto PDA, TYGA, and TG media, respectively; the cultures were kept at 28 °C and the diameter of the fungal colony was recorded daily for 6 consecutive days. Then, the WT and mutant strains were inoculated separately onto CMY medium and cultured at 28 °C for 5 days. The cultured WT and mutant strains were divided into two groups. From the first group, a conidia suspension was prepared with sterile water, mixed thoroughly, and a 20 μL aliquot was taken for counting. From the second group, conidia were cut down where they grew evenly and were placed on their side under an inverted fluorescence microscope for observation.

### 2.7. Formation of Traps and Measurement

To determine the activity of ammonia and ascaroside (Acsr#18) on *A. oligospora* producing traps, spores of *A. oligospora* were washed off from 7-day cultures using sterile water. From the resulting suspension, a 20 μL aliquot was obtained to count the number of spores. Then, 3 × 10^3^ spores were transferred onto three 6 cm plates containing WA medium; the plates were sealed and incubated at 28 °C for 36 h, after which 1 mL of inducers (nitrogen signals: ammonia, urea, NH_4_Cl, and glutamine; carbon signal: Ascr#18.) was added to the plates. The inducers used were ammonia (Am) (25% ammonia diluted 10^3^ times), Acsr#18 (10 nM, artificial from MedChemExpress, State of New Jersey, USA), and the mixture of ammonia and Acsr#18. The condition of the *A. oligospora* trap production was observed and recorded according to the time gradient of 24 h, 48 h, and 72 h [39]. Each experiment was repeated 3 times.

### 2.8. Statistical Analyses

Graphing and data analysis were performed using GraphPad Prism version 9.5.1 (GraphPad Software, San Diego, CA, USA). The statistical significance of variables was measured using the *t*-test for pairwise comparisons of means, and experimental data were expressed as the standard deviation (SD) of the mean. A value of *p* < 0.05 was considered statistically significant, and vice versa. Statistical significance was defined at * *p* < 0.05, ** *p* < 0.01, and *** *p* < 0.001.

## 3. Results

### 3.1. Effect of Am and Ascr#18 on A. oligospora Trap Formation

Ammonia (Am) and Acsr#18 are important triggers in the morphological conversion in *A. oligospora*. The results showed that Ascr#18 or Am alone induced a number of traps. The mixture of Am plus Ascr#18 significantly increased trap numbers (Figure 1A,B). Consistently, other nitrogenous inducers like urea, NH_4_Cl, and glutamine (Glu) also displayed synergy in trap formation (Figure 1C). These data revealed that Ascr#18 plus Am promoted higher trap formation in *A. oligospora*.

### 3.2. Amt43 Is Ammonia Transporter of A. oligospora

There are three annotated ammonia permeases from the *A. oligospora* genome (https://www.ncbi.nlm.nih.gov/bioproject/41495, accessed on 18 July 2022), named as Amt06 (AOL_s00006g212p), Amt43 (AOL_s00043g163p), and Amt80 (AOL_s00080g393p). These ammonia transporters exhibit significant similarity in their major functional domain of ammonium transporter Amt B-like (Figure 2A). Among them, Amt43 represents the highest conservation with the model yeast *Saccharomyces cerevisiae* Mep2 [40], which has been referred to as the most important ammonia transceptor due to its highest affinity to ammonia/ammonium (Figure 2B). In addition, structural modeling revealed that the Amt43 transporter consists of 11 helices embedded in the cell membrane and is probably responsible for the transport of ammonium ions or ammonia (Figure 2C, Appendix A).

### 3.3. Ascr#18 Interacts with Am in Trap Induction

In the GFP-labeled strain (Amt43::GFP), Amt43 was mostly expressed on the plasma membrane (Figure 3A,B). When exposed to Am solution (13.3 mM), the GFP signal was immediately translocated to the cytoplasm within 5 min, which characterizes endocytosis. However, Ascr#18 alone did not change the Amt43 GFP localization. The mixture of two inducing factors weakened the Amt43 GFP internalization and alleviated the abnormality of the vacuolar phenotype (Figure 3C). Moreover, the endocytic marker FW4-64 analyses showed that Am produced obvious endocytosis within 1 min, while Ascr#18 exposure also caused slight endocytosis within 5 min. The mix of Am and Ascr#18 facilitated endocytosis and maintained a relatively intact intracellular state in the fungal cell (Figure 3D).

### 3.4. Am Can Facilitate Ascr#18 in Inducing A. oligospora to Produce Traps

To determine whether Amt43 is linked to trap formation, the Amt43 coding gene (Aol_s00043g163p) was deleted and further complemented using a homologous complement and the corresponding mutations were verified using Southern blot analysis (Appendix A). By contrast, the growth and conidiation of the mutants (Δ*AoAmt43*-1, Δ*AoAmt43*-2, and Δ*AoAmt43*-3) were partially blocked (Figure 4A,B; Appendix A). Further, the exposure to four chemicals (Am, 25% ammonia diluted 10^3^ times; NH_4_Cl, 37.4 mM; urea, 33.3 mM; Glu, 3.4 mM) led to fewer traps in Δ*AoAmt43* mutants against the WT after 72 h of treatment; conversely, the Amt-rescued counterparts were paralleled to that of the WT phenotype (Figure 4C). Subsequently, it showed that the Amt43 mutation did not affect the trap production induced by Ascr#18. But the extra addition of Ascr#18 increased the trap number more in contrast to ammonia alone to the Amt mutant, indicating that the ammonia channel is not involved in Ascr#18 uptake (Figure 4D).

## 4. Discussion

The specialization of nutritive mycelium into traps is a survival strategy for the environmental adaptation of NT fungi [42]. Multiple cues mediate the morphological conversion from the saprophytic to parasitic state, such as nematodes, ascarosides, urea, and ammonia. Typically, *A. oligospora* forms 3D adhesive nets stimulated by these inducers to capture and digest nematode prey for nutrition and survival [35,43]. According to this study, traps can be induced by a combination of nitrogenous cues and ascarosides, and ammonia and Ascr#18 synergistically regulate the induction of the adhesive net of *A. oligospora*, which may provide a dominant trapping population for NT fungi in complex soil habitats.

The fungal lifestyle transition is a responsive strategy to obtain nutrients like carbon sources or nitrogen sources. This is so that carbonaceous or nitrogenous substances are conservatively obligated to turn to the inducing signal for the predatory structure [27]. Earlier studies found that urea, ammonia, nematodes, or ascaroside alone could trigger trap formation. The mix of carbonaceous and nitrogenous chemicals used in this study has shown a significant effect on trap production in different concentrations rather than a simple additive effect. Of all the mixes tested, Ascr#18 plus ammonia promoted more trap formation in *A. oligospora* than the rest (Figure 1). Plants prefer ammonia/ammonium as a nitrogen source [44], but NT fungi have evolved to respond to this small molecule signal to change their lifestyle in response to it. The underlying mechanism of ammonia-mediated traps may be attributed to cellular endocytosis, because the FM4-64 fluorescent dye has marked the obvious plasma internalization and the massive endosome enriched during the induction process of adhesive nets in *A. oligospora* (Figure 3D). The effect of ammonia/ammonium on endocytosis has been widely found in eukaryotes, especially in yeast and plant cells [45,46]. This study showed that the induction of ammonia mixed with Ascr#18 provides a relatively intact intracellular state in the fungal cell away from endocytosis, suggesting that Ascr#18 and ammonia can crosstalk for trap development in NT fungi.

Ammonia, ammonium, urea, and glutamine are common nitrogenous substances for trap induction in NT fungi. Am is a typical preferred nitrogen source and the most potent signal in the trap induction process, compared to urea and NH_4_Cl, urea, and glutamine. Furthermore, ammonia is the final product of the nitrogen cycle. Therefore, we use ammonia as the main nitrogen signal for the discussion and experiments in this paper. The ammonia transporter is a main channel for ammonia uptake. There are three annotated ammonia permeases from the *A. oligospora* genome, namely, Amt06 (AOL_s00006g212p), Amt43 (AOL_s00043g163p), and Amt80 (AOL_s00080g393p). Among them, Amt43 is highly homologous with yeast ammonium transcepter Mep2, which is responsible for the sense and transportation of ammonia/ammonium, and consequently is involved in the regulation of pseudohyphal growth [47]. It was found in the present study that Amt43 is associated with the morphological conversion in *A. oligospora.* Ammonia plus ascaroside can promote higher trap numbers; however, the ammonia channel is not involved in Ascr#18 uptake. Rather, ammonia is involved in Ascr#18-induced trap formation (Figure 4D).

Ascaroside is a prominent signal group in nematodes, which controls developmental diapause and social behaviors like aggregation and mate finding in nematode species [48,49]. However, fungi become nematode predators by acquiring the conserved ascarosides released by nematodes. Many ascarosides have been identified as inducers of trap structures, such as Ascr#5, #7, and #18 [50]. The addition of Ascr#18 in this study did indeed promote the production of adhesive nets in *A. oligospora* (Figure 1B). From the up-regulation of trap yield in biocontrol to nematode disease, this work shows the combination analysis of different signals. In addition to the synergistic effect of Ascr#18 and ammonia, Ascr#18 also exhibited an important super effect with ammonium, urea, or glutamate (Figure 1C). Therefore, multiple factors mediate the trap induction in the soil context.

Over 200 ascarosides have been identified in the nematode phylum, and NT fungi may use protein signaling to interact with these compounds [51]. The ammonia transceptor Amt43 was analyzed to seek the underlying mechanism for the synergistic effect on trap formation (Figure 4B). Confirmatively, ammonia induces adhesive nets through PM endocytosis based on the GFP observation of Amt43 and FM4-64 tracks. Even though Ascr#18 treatment can cause mild endocytosis, it can compensate or alleviate the cell damage caused by ammonia to a certain extent, indicating that ascaroside compounds are beneficial for organisms to resist external adverse conditions, whether for the nematode itself or its natural enemies. It was previously found that conserved nematode signaling molecules like Ascr#18 elicit plant defenses and pathogen resistance against bacterial, fungal, and nematode infection [52].

In summary, this work demonstrates that the NT fungi can establish trap structures by nitrogenous signals and carbonaceous signals in a synergistic manner (Figure 5). In *A. oligospora*, the transporter is responsible for ammonia perception and regulates the formation of adhesive nets depending on endocytosis; the extra combination of conserved ascarosides provides a better fit for the fungus against ammonia stress. As a result of this study, we gain new insight into the regulation of multiple signals in traps for biocontrol of nematode pests.

## Figures and Tables

**Figure 1 pathogens-12-01114-f001:**
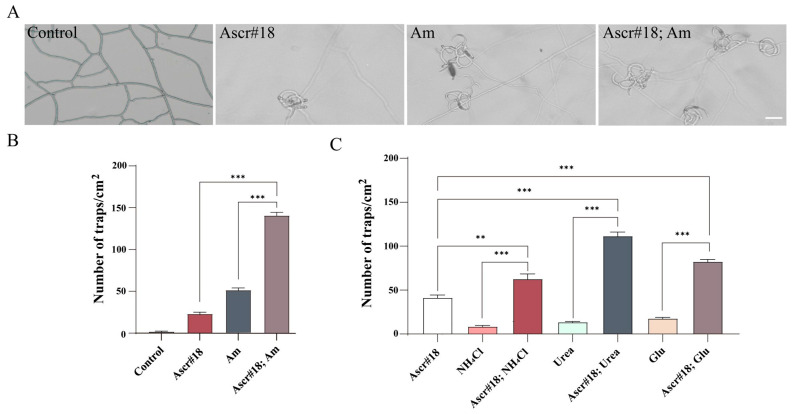
Measurement of trap-inducing activity by nitrogenous and ascaroside signals. (**A**) Comparison of adhesive traps produced by the overlay effect of Ascr#18 and Am (Bar = 100 μm). (**B**) The mix of Ascr#18 and Am significantly promoted the formation of traps. (**C**) Comparison of traps produced by the overlay effect of Ascr#18 and the different nitrogen signals; ** *p* < 0.01, *** *p* < 0.001.

**Figure 2 pathogens-12-01114-f002:**
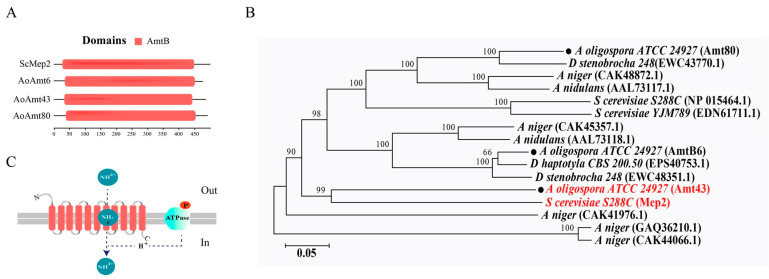
Domain analyses of ammonia transport proteins. (**A**) Comparison of gene functional domains between *Saccharomyces cerevisiae* Mep2 and *Arthrobotrys oligospora* AmtBs (AoAmt6, AoAmt43, AoAmt80). The numerical scale shows the length of amino acid chains; it was presented by NCBI and TBtools software ((version 1.09861, CJ chen, Guangdong Province, China). (**B**) The phylogenetic tree of Amt in fungi (*Arthrobotrys oligospora*, *Drechslerella stenobrocha*, *Dactylellina haptotyla, Aspergillus niger*, *Aspergillus nidulans*, *Saccharomyces cerevisiae*); the tree was constructed using Mega 7 software (version 7.0, Mega Limited, Auckland, New Zealand). (**C**) The 11 helices of Amt43 and the ammonia transport process in fungi, referenced from “Hypothetical transport mechanisms of Mep/Amt proteins” [41].

**Figure 3 pathogens-12-01114-f003:**
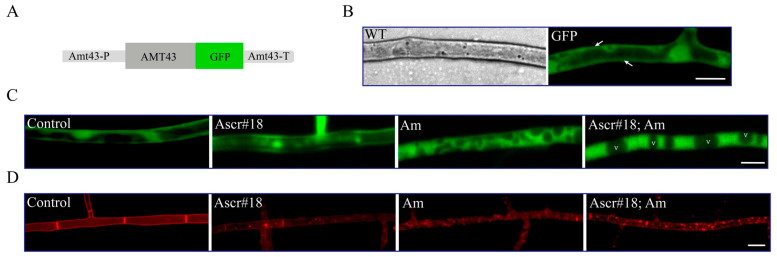
The dynamic response of *A. oligospora* by the addition of Ascr#18 or/and Am. (**A**) Schematic diagram of Amt43::GFP fluorescent labeling. (**B**) Green fluorescence shows that Amt43 is mainly localized on the cell membrane of the hyphal of the NT fungus *A. oligospora* (bar = 10 μm). (**C**) Amt43::GFP dynamics responding to Ascr#18 or/and Am (bar = 10 μm), v is for vacuole. (**D**) Dynamics response of endocytosis by Ascr#18 or/and Am (bar = 10 μm).

**Figure 4 pathogens-12-01114-f004:**
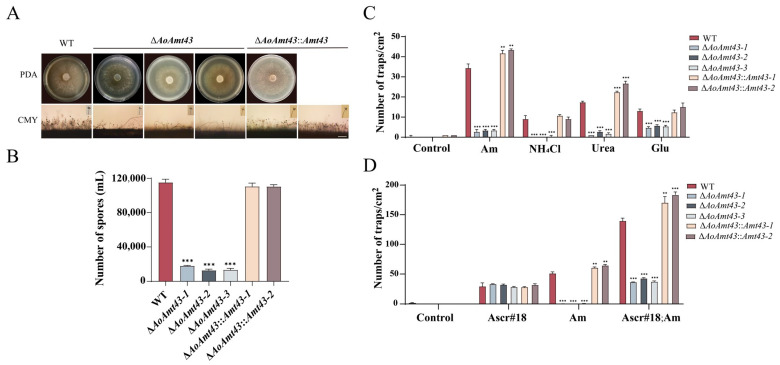
The growth and trap-inducing activity in strains of the WT and Δ*AoAmt43*. (**A**) Measuring the growth and spore number of Δ*AoAmt43* (PDA bar = 1 cm, CMY bar = 100 μm). (**B**) Comparison of the spore numbers of WT, Δ*AoAmt43*, and Δ*AoAmt43::Amt43*. (**C**) The trap-inducing activity of Δ*AoAmt43* under the different nitrogen signals. Am, 25% ammonia diluted 10^3^ times; NH_4_Cl, 37.4 mM; urea, 33.3 mM; Glu (glutamine), 3.4 mM. (**D**) The trap-inducing activity of Δ*AoAmt43* under ammonia or/and Ascr#18 (Am, 25% ammonia diluted 10^3^ times; Ascr#18, 10 nM); unpaired *t*-test, parametric test, ** *p* < 0.01, *** *p* < 0.001.

**Figure 5 pathogens-12-01114-f005:**
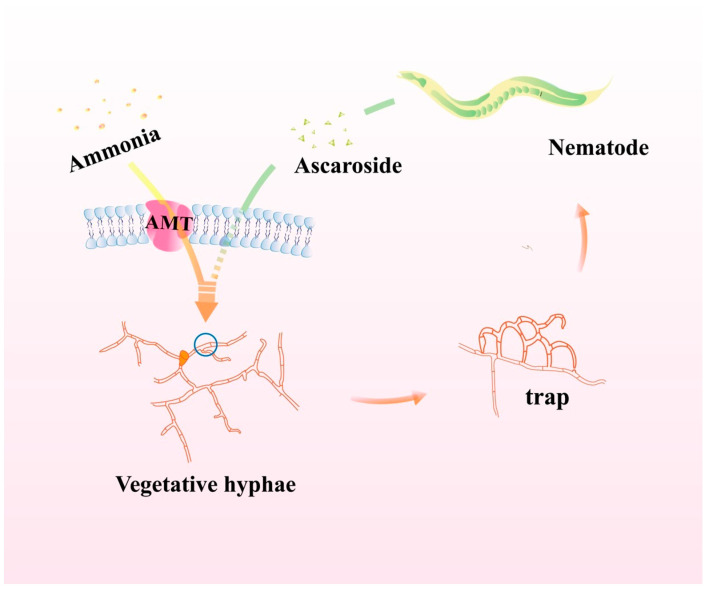
The model of ammonia and Ascr#18 on trap formation in *A. oligospora*. Entry of exogenous ammonia into *A. oligospora* mycelial cells via ammonia transport proteins in the cell membrane, and synergism with exogenous ascarosides in the cell, promotes the formation of adhesive nets of traps by *A. oligospora*, which in turn traps and kills nematodes more rapidly and controls nematode populations.

## Data Availability

The original contributions presented in the study are included in the article/Supplementary Material, and further inquiries can be directed to the corresponding author.

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
