# Peer review of "Ammonia and Nematode Ascaroside Are Synergistic in Trap Formation in Arthrobotrys oligospora"

_pathogens, 2023, doi:10.3390/pathogens12091114_

Round 1
Reviewer 1 Report
This manuscript is highly relevant to the field of nematophagous fungi; it has been well designed and merits publication.
The manuscript, though, needs a lot of work before is published. The authors are highly encour need to follow the recommendations done. The English language must be improved, as it is at present it hinders seriously the quality of the manuscript. The sections Materials and Methods and Results needs serious improvement. Interestingly, given the quality of the English usage, the paper seems to be divided in two major sections, i.e. a) Introduction and Discussion are well written and presented; b) Materials and Methods and Results are poorly written.
I look forward to read the improved version of this manuscript.

Author Response
Dear reviewer,
Many thanks for your logical comments on our paper. As a whole, the comments are encouraging and constructive. We have revised the whole manuscript very carefully according to your valuable comments that are as follows:
Comment The sections Materials and Methods and Results needs serious improvement.
Answer Thank you very much for your suggestion. In the Materials and Methods and Results sections, we have added several descriptions based on your suggestions.
- Materials and Methods (125-140): Positive transformants were analyzed by Southern Blot. The hybridization probe was amplified using primers Amt43ko-F/Amt43ko-R, and restriction enzyme MefI was used to digest the genomic DNA of oligospora, and experiments were carried out according to the instructions provided in the North2South® Chemiluminescent Hybridisation and Detection Kit (Pierce, Rockford, USA). Subsequently, the samples were washed with 0.25 M HCl for 13-15 min, rinsed with ddH2O for 15 min (repeated 3 times), and treated with denaturing solution (NaOH 0.5 M/L, NaCl 1.5 M/L) for 15 min (repeated 2 times), rinsed in ddH2O for 15 min, treated with neutralizing solution (Tris-HCl 0.5 M/L, NaCl 1.5 M/L, PH=7.5) for 15 min (repeated 2 times), rinsed in ddH2O for 15 min and equilibrated in 20X SSC for 10 min, while the nylon membrane was soaked in ddH2O for 5 min and then transferred to equilibrate in 20X SSC for 30 min, salt bridges were constructed for transferring membranes for more than 30 h. After the membrane transfer, the nylon membrane was cleaned with 6X SSC, blown dry with cold air and then cross-linked with 1200 μJ/cm2 UV, and hybridization, membrane washing and imaging finally confirmed that the positive transformants obtained were the correct mutants.
- Materials and Methods (149-152): After the colonies to grow for 7 days, they were transferred to WA medium with cellophane on the surface and incubated at 28°C for 3-4 days, and the cellophane with mycelium areas (0.5x0.5 cm) were taken and placed under the fluorescence microscope to observe whether the mycelium could emit fluorescence.
- Materials and Methods (164-172): To assess the growth differences between WT and mutant strains, the two strains were first cultured for adaptation on PDA medium for 7 days at 28°C. Then, we collected the same size (6x6 mm) blocks and inoculated them onto PDA, TYGA and TG media, respectively, the cultures were kept at 28°C and the diameter of the fungal colony was recorded during 6 consecutive days. We then inoculated the WT and mutant strains separately onto CMY medium and cultured them at 28°C for 5 days. The cultured WT and mutant strains were divided into two groups. From the first group, conidia suspension was prepared with sterile water, mixed thoroughly and a 20μL aliquot was taken for counting. From the second group, conidia were cut down the conidia where it grew evenly and placed it on its side under an inverted fluorescence microscope for observation.
- Materials and Methods (174-182): Spores of oligospora were washed off from 7 days-cultures using sterile water. From the resulting suspension a 20 μL aliquot was obtained to count the number of spores. Then 3x103 spores were transferred onto 6 cm plates containing WA medium, the plates were sealed and incubated at 28°C for 36 h, after which 1 mL of inducers were added to the plates. The inducers used were ammonia (Am) (25% ammonia diluted 103 times), Acsr#18 (10 nM, artificial from MCE), and the mixture of ammonia and Acsr#18. The condition of the A. oligospora trap-production was observed and recorded according to the time gradient of 24 h, 48 h and 72 h. Each experiment was repeated 3 times.
- Results (210-212): There are 3 annotated ammonia permeases from oligospora genome (https://www.ncbi.nlm.nih.gov/bioproject/41495), named as Amt06 (AOL_s00006g212p), Amt43 (AOL_s00043g163p), Amt80 (AOL_s00080g393p).
- Results (252-255): Further, the exposure to four chemicals (Am, 25% ammonia diluted 103 times; NH4Cl, 37.4 mM; Urea, 33.3 mM; Glu, 3.4 mM) led to fewer traps in ΔAoAmt43 mutants against the WT after 72 h treatment, conversely, the Amt rescued counterpart were paralleled to that of the WT phenotype.
"Please see the attachment."
Sincerely wishing you the best!
Yours sincerely,
Keqin Zhang

Reviewer 2 Report
Ammonia and nematode ascaroside are synergistic on trap formation in Arthrobotrys oligospora
The conclusions of the work were: Therefore, there is a synergistic effect on trap induction from different nitrogenous and ascaroside signals.
The abstract is very brief, but sufficient within the journal's standards.
Keywords should be different from the title.
In the introduction or even in the discussion the role of amino acids, nanoparticles, enzymes and seroproteases was missing and for this we recommend to visualise some reviews:
https://doi.org/10.3390/parasitologia1030018
https://link.springer.com/article/10.1007/s00253-013-5366-z
The methodology is adequate and is well described for the most part, however, it should detail a little more how: Line 140 After waiting for the colonies to grow well, they were moved to a fluorescent microscope to observe whether the mycelium could fluoresce.
Time? observation in what range? and others. Do a full review.
The results are well consistent and presented.
The discussion should follow the order of presenting the results and then discussing them according to the literature and not presenting the literature first and then putting the results accordingly and as an example in the first paragraph. Line 253.
Only revisions to paragraph descriptions should be avoided.
The wording should be in an impersonal form as in line 304.
Author Response
Dear reviewer ,
Many thanks for your constructive comments on our paper. We have revised the whole manuscript very carefully according to your valuable comments. Thank you very much again.
Comment Keywords should be different from the title. In the introduction or even in the discussion the role of amino acids, nanoparticles, enzymes and seroproteases was missing and for this we recommend to visualise some reviews. The methodology is adequate and is well described for the most part, however, it should detail a little more how: Line 140 After waiting for the colonies to grow well, they were moved to a fluorescent microscope to observe whether the mycelium could fluoresce. Time? observation in what range? and others. Do a full review. The results are well consistent and presented. The discussion should follow the order of presenting the results and then discussing them according to the literature and not presenting the literature first and then putting the results accordingly and as an example in the first paragraph. Line 253. Only revisions to paragraph descriptions should be avoided. The wording should be in an impersonal form as in line 304.
Answer Thank you very much for your suggestion. Here are our answers.
- Keywords (23): Arthrobotrys oligospora; ammonia; Acsr#18; Amt43; trap formation; synergy.
- Materials and Methods (125-140): Positive transformants were analyzed by Southern Blot. The hybridization probe was amplified using primers Amt43ko-F/Amt43ko-R, and restriction enzyme MefI was used to digest the genomic DNA of oligospora, and experiments were carried out according to the instructions provided in the North2South® Chemiluminescent Hybridisation and Detection Kit (Pierce, Rockford, USA). Subsequently, the samples were washed with 0.25 M HCl for 13-15 min, rinsed with ddH2O for 15 min (repeated 3 times), and treated with denaturing solution (NaOH 0.5 M/L, NaCl 1.5 M/L) for 15 min (repeated 2 times), rinsed in ddH2O for 15 min, treated with neutralizing solution (Tris-HCl 0.5 M/L, NaCl 1.5 M/L, PH=7.5) for 15 min (repeated 2 times), rinsed in ddH2O for 15 min and equilibrated in 20X SSC for 10 min, while the nylon membrane was soaked in ddH2O for 5 min and then transferred to equilibrate in 20X SSC for 30 min, salt bridges were constructed for transferring membranes for more than 30 h. After the membrane transfer, the nylon membrane was cleaned with 6X SSC, blown dry with cold air and then cross-linked with 1200 μJ/cm2 UV, and hybridization, membrane washing and imaging finally confirmed that the positive transformants obtained were the correct mutants.
- Materials and Methods (149-152): After the colonies to grow for 7 days, they were transferred to WA medium with cellophane on the surface and incubated at 28°C for 3-4 days, and the cellophane with mycelium areas (0.5x0.5 cm) were taken and placed under the fluorescence microscope to observe whether the mycelium could emit fluorescence.
- Materials and Methods (164-172): To assess the growth differences between WT and mutant strains, the two strains were first cultured for adaptation on PDA medium for 7 days at 28°C. Then, we collected the same size (6x6 mm) blocks and inoculated them onto PDA, TYGA and TG media, respectively, the cultures were kept at 28°C and the diameter of the fungal colony was recorded during 6 consecutive days. We then inoculated the WT and mutant strains separately onto CMY medium and cultured them at 28°C for 5 days. The cultured WT and mutant strains were divided into two groups. From the first group, conidia suspension was prepared with sterile water, mixed thoroughly and a 20μL aliquot was taken for counting. From the second group, conidia were cut down the conidia where it grew evenly and placed it on its side under an inverted fluorescence microscope for observation.
- Materials and Methods (174-182): Spores of oligospora were washed off from 7 days-cultures using sterile water. From the resulting suspension a 20 μL aliquot was obtained to count the number of spores. Then 3x103 spores were transferred onto 6 cm plates containing WA medium, the plates were sealed and incubated at 28°C for 36 h, after which 1 mL of inducers were added to the plates. The inducers used were ammonia (Am) (25% ammonia diluted 103 times), Acsr#18 (10 nM, artificial from MCE), and the mixture of ammonia and Acsr#18. The condition of the A. oligospora trap-production was observed and recorded according to the time gradient of 24 h, 48 h and 72 h. Each experiment was repeated 3 times.
- Discussion (268-275): Specialization of nutritive mycelium into traps as a survival strategy for environmental adaptation of NT fungi. Multiple cues mediate the morphological conversion from saprophytic to parasitic, such as nematodes, ascarosides, urea and ammonia. Typically, oligospora forms 3D adhesive nets stimulated by these inducers to capture and digest nematode prey for nutrition and survival. According to this study, traps can be induced by a combination of nitrogeneous cues and ascarosides, and ammonia and Ascr#18 synergistically regulate the induction of the adhesive net of A. oligospora, which may provide a dominant trapping population for NT fungi in complex soil habitats.
- Discussion (295-296): There are 3 annotated ammonia permeases from oligospora genome, named as Amt06 (AOL_s00006g212p), Amt43 (AOL_s00043g163p), Amt80 (AOL_s00080g393p).
- Discussion (300-302): Ammonia plus ascaroside can promote more trap numbers, however, ammonia channel is not involved in Ascr#18 uptake; rather than in Ascr#18-induced trap formation.
- References (418-419, 438-439): Add two references (Recent Advances in the Control of Helminths of Domestic Animals by Helminthophagous Fungi; Nematophagous fungi for biological control of gastrointestinal nematodes in domestic animals.).
"Please see the attachment."
Sincerely wishing you the best!
Yours sincerely,
Keqin Zhang

Reviewer 3 Report
Thank you for the opportunity to review this paper. I have thoroughly read the manuscript and provided minor comments that need to be addressed. Overall, I believe the paper is well-written, and the experiment is valid and properly designed and analyzed.
Furthermore, I recommend that the authors reorganize the discussion section. The obtained results in this research are valuable and interesting, but they are challenging to understand in the current form of the discussion. My suggestion is to first explain the most significant findings of this experiment and then compare them with existing research.
Finally see the pdf file

Author Response
Dear reviewer,
Many thanks for your constructive comments on our paper. We have revised the whole manuscript very carefully according to your valuable comments. Thank you very much again.
Comment Recommend that the authors reorganize the discussion section. The obtained results in this research are valuable and interesting, but they are challenging to understand in the current form of the discussion.
Answer Thank you very much for your suggestion. Here are the changes we made to the Discussion section.
- Discussion (268-275): Specialization of nutritive mycelium into traps as a survival strategy for environmental adaptation of NT fungi. Multiple cues mediate the morphological conversion from saprophytic to parasitic, such as nematodes, ascarosides, urea and ammonia. Typically, oligospora forms 3D adhesive nets stimulated by these inducers to capture and digest nematode prey for nutrition and survival. According to this study, traps can be induced by a combination of nitrogeneous cues and ascarosides, and ammonia and Ascr#18 synergistically regulate the induction of the adhesive net of A. oligospora, which may provide a dominant trapping population for NT fungi in complex soil habitats.
- Discussion (295-296): There are 3 annotated ammonia permeases from oligospora genome, named as Amt06 (AOL_s00006g212p), Amt43 (AOL_s00043g163p), Amt80 (AOL_s00080g393p).
- Discussion (300-302): Ammonia plus ascaroside can promote more trap numbers, however, ammonia channel is not involved in Ascr#18 uptake; rather than in Ascr#18-induced trap formation.
"Please see the attachment."
Sincerely wishing you the best!
Yours sincerely,
Keqin Zhang

Round 2
Reviewer 1 Report
The manuscript has improved tremendously since the first version. It is not ready for publication, though. Some further improvements are needed in relation to explanation/description of M&M (e.g. the lack of explanation on using other nitrogenous inducers other than ammonia and Ascr#18 is of particular concern), and language use/semantics . Some of the comments had already been made on the previous version but have not been addressed.
Figures must be improved as quality is lost during the files export from GraphPad to MS Word, especifically:
Figure 1C, 1D and Figure 4B, 4C and 4D: the quality and size of axis and series legends must be improved as it is barely readable on a digital copy at zoom 125%. If any reader decides to print the paper the figures will be impossible to understand.
Figure 4 C and D: Delete all the "ns" legends over the bars, leave only the significant asterisks.
Figure 5 is pretty much a repetition of Fig 1A, it shows excatly the same concept. More comments in the file itself.
The rest of the comments and corrections are in the manuscript itself.
Looking forward to the ready-to-publish version of the manuscript!
It must be improved, as the comments on this second reviewed version show. On the revised version the authors just replaced the indicated text with the one supplied by this reviewer, but did not check for language use and quality throughout the rest of the manuscript.
Author Response
Dear reviewer,
Thank you very much for your concerns and suggestions. Your feedback has been very helpful and we have made changes in response to your suggestions.
Comment: The lack of explanation for using other nitrogenous inducers than ammonia and Ascr#18 is of particular concern.
Figures must be improved as quality is lost during the files export from GraphPad to MS Word, specifically Figure 1C, 1D and Figure 4B, 4C and 4D: the quality and size of axis and series legends must be improved as it is barely readable on a digital copy at zoom 125%. If any reader decides to print the paper the figures will be impossible to understand.
Figure 4 C and D: Delete all the "ns" legends over the bars, leave only the significant asterisks. Figure 5 is pretty much a repetition of Fig 1A, it shows exactly the same concept.
Answer: we appreciate your suggestions, and we'll answer you in points.
- The synergistic effect of other nitrogenous compounds with Ascr #18 is rarely discussed in this paper. The reason for this is that we have demonstrated through experiments that the synergistically induced uptake of ammonia and Ascr #18 is significantly better than other nitrogen sources. Ammonia is a typical preferred nitrogen source and the most potent signal in trap induction process, compared to urea and NH4Cl, urea, and glutamine. Furthermore, ammonia is the final product of the nitrogen cycle. Therefore, we use ammonia as the main nitrogen signal for the discussion and experiments in this paper.
- In response to your suggestion about the quality of the images we have made changes to the revised manuscript. Please see attached.
- The "ns" in Figure 4 C and D has been removed.
- Figure 5 overlaps Figure 1A, so, we deleted the Figure 1A.
Yours sincerely,
Keqin Zhang
